# Obesity and Body Composition in Relation to Liver and Kidney Ultrasound Elastography in Paediatric Patients with Either Hypertension or Chronic Kidney Disease

**DOI:** 10.3390/children11010018

**Published:** 2023-12-23

**Authors:** Nataša Marčun Varda, Sonja Golob Jančič, Mirjam Močnik

**Affiliations:** 1Department of Paediatrics, University Medical Centre Maribor, Ljubljanska 5, 2000 Maribor, Slovenia; sonja.golobjancic@ukc-mb.si (S.G.J.); mirjam.mocnik@ukc-mb.si (M.M.); 2Medical Faculty, University of Maribor, Taborska 8, 2000 Maribor, Slovenia

**Keywords:** ultrasound elastography, body composition, children, fat mass, liver, kidney

## Abstract

(1) Background: Ultrasound elastography is a novel ultrasound technique for evaluating tissue elasticity. One of the key factors influencing the measurement in children is excess weight. This study aimed to evaluate the effect of body composition, namely, fat mass, on liver and kidney ultrasound elastography in paediatric patients. (2) Methods: 114 participants, in whom bioimpedance, along with liver and kidney ultrasound elastography, were performed, were included (37 patients with chronic kidney disease, 46 patients with hypertension, and 31 healthy subjects). (3) Results: Bioimpedance analysis showed a significant correlation between liver elastography parameters and the phase angle (*p* = 0.002), fat-free mass (*p* = 0.001), body cell mass (*p* = 0.001), total body water (*p* = 0.001), extracellular water (*p* = 0.006), and, to lesser extent, fat mass (*p* = 0.041). On the contrary, kidney elastography parameters strongly correlated only with fat mass (*p* < 0.001 for both kidneys). (4) Conclusions: Liver and kidney stiffness increased in overweight participants and showed significant correlation with fat mass, particularly in the case of kidney elastography.

## 1. Introduction

Ultrasound elastography (UE) is a newly researched method which can be applied in several fields of adult and paediatric medicine [1,2]. By using ultrasound, it estimates tissue stiffness as a physical property through monitoring tissue movement as energy is applied. Tissue stiffness is known to be a marker of tissue pathology, which affects UE, along with patient- and ultrasound-operator-associated factors [3]. 

Weight status, along with other patient-based factors, significantly affects UE measurement in children [4]. Liver stiffness is strongly related with body mass index (BMI) in children with obesity. It was higher in participants with an abnormal echogenic liver [5], thus showing potential as a tool for the early detection of paediatric metabolic-dysfunction-associated fatty liver disease (MAFLD) [6,7]. Obesity seems to have a higher impact on liver stiffness regardless of other factors, such as metabolic syndrome and insulin resistance [8]. In addition, reduced weight resulted in an increase of liver elasticity in children with MAFLD [9]. Interestingly, in some studies, where liver fat mass was measured using established imaging and histological diagnostics, liver stiffness did not correlate with the quantification of fat mass [10]. Similarly, there was no proven association between elasticity and kidney fibrosis in children when comparing kidney elastography and histology [11]; however, renal cortical elasticity was decreased in overweight children [12]. 

BMI, a commonly used measure of overweight and adiposity, lacks a reliable assessment of body composition (BC), namely, fat mass, which contributes to complications of obesity. The exact measurement of fat mass along with other BC parameters is important in the management of obese children to accurately assess the gravity of adiposity, its impact on the pathophysiology of the disease, and the effectiveness of interventions [13]. Bioelectrical impedance analysis (BIA) is a commonly used method to assess BC. It evaluates it through the impedance of a small electric current that travels through the body with different speeds. A low-voltage electric current is applied on one extremity. The time difference in electric current at the sensing electrode enables the evaluation of the BC due to the difference in resistance in various tissues [14]. It also allows a fast and easy monitoring of the evolution of the nutritional and hydration state [14]. 

To additionally explore the relationship between obesity and UE, body impedance measurement and UE were performed, and the relationship between both analysed. The emphasis was on fat mass evaluation. The aim of this study was, therefore, to evaluate the effect of BC and fat mass on liver and kidney UE in paediatric patients with either hypertension (HTN) or chronic kidney disease (CKD) compared to healthy participants. We hypothesise that both liver and kidney elastography will correlate significantly with fat mass, especially liver elastography, that was already associated with MAFLD in earlier studies.

HTN and CKD are important cardiovascular risk factors that significantly contribute to cardiovascular burden, leading to increased morbidity and mortality in adult life. The rate of both HTN and CKD increase with obesity. Therefore, the early detection of cardiovascular- and obesity-associated changes is imperative in paediatric high-risk patients to secure appropriate monitoring and management. In this study, BIA and UE were used as test tools to seek additional insight in children with increased cardiovascular- and obesity-associated risk.

## 2. Materials and Methods

### 2.1. Subjects

This study is an addition to previously published study about use of liver and kidney elastography in paediatric patients with HTN or CKD, where patients’ selection and elastography methods are described in detail [15]. Briefly, patients with CKD or essential HTN were recruited during their admittance for routine follow-up and additional diagnostic procedures. Healthy subjects were invited separately. In all participants, liver and kidney UE were performed with already published results [15]. The majority of the participants had body impedance measurement performed during their hospital follow-up. Finally, 114 of 129 participants were included in current study: 37 patients with CKD stage 1 or 2 (Group 1)—patients with mildly affected kidney function, 46 patients with mild HTN (Group 2), and 31 healthy participants to provide comparison. The diagnosis of HTN was established in accordance with diagnostic guidelines [16]. In the group with healthy participants, overweight or obese children were excluded, as well as children with any chronic condition. 

The study was conducted in accordance with known ethical principles and approved by the Institutional and National Ethics Committee (protocol codes UKC-MB-KME-35/20 and 0120-372/2020/6, respectively). A written consent form was signed either by participating young adults themselves or by parents/legal guardians of under-aged participants.

### 2.2. Investigations

Anthropometric measurements were carried out on all subjects. In the second part of the study, all participants were divided into overweight/obese and normal-weight group according to international limits for BMI above 85th percentile [17]. Altogether, this study included 43 overweight and 71 normal-weight children and young adults across all groups. Blood pressure was measured on all participants using an oscillometric method (Omron Healthcare Co., Muko, Japan).

Next, BC was measured using BIA (Nutrilab Bioimpedance, Akern 2016) following manufacturer’s instructions. The measurement was performed in fasted state, with an empty bladder, in the same lying position with the same electrodes position. In addition, the environment was always the same with hospital room temperature and no jewelry. Patients were required to avoid alcohol and caffeine consumption and to avoid exercise for 12 h before the measurement. BIA gave the results of each child’s BC, namely, phase angle (PA), fat mass (FM), fat-free mass (FFM), body cell mass (BCM), total body water (TBW), and extracellular water (ECW).

Finally, ultrasound elastography was performed in all participants as described in previous publication [15]. Liver and kidney ultrasound elastography were performed using Canon aplio-a ultrasound and 2D shear wave elastography according to manufacturer’s instructions. The quality of measurement was achieved with standard deviation (SD) < 20% of mean in designated area. Ten areas in cortical region of each kidney and thirteen areas in the liver were measured. The median value of the measurements was computed with ultrasound’s software [15]. 

### 2.3. Statistical Analysis

Statistical analyses were carried out using SPSS Statistics (IBM, version 22). The patients’ cohort was described using descriptive statistics. The normality of the parameters was assessed using Kolmogorov–Smirnov and Shapiro–Wilk tests. The variables were not normally distributed and are presented as medians (interquartile range), where appropriate. Appropriately, nonparametric tests were used (Mann–Whitney and Kruskal–Wallis test, and Spearman correlation coefficient). A multiple regression analysis was performed to evaluate potential confounders among available variables. A value of *p* < 0.05 was considered statistically significant.

## 3. Results

Groups are comparable according to age and height (Table 1); however, they significantly differ in weight and related anthropometric parameters, as well as systolic pressure. All groups differ in both liver and kidney elastography and in BC. Fat mass was significantly higher in Group 2, where obesity-related HTN predominates. Liver stiffness was increased in both groups with patients included and was further aggravated after obesity-status division. There was no difference in kidney elastography between patients and healthy individuals; however, kidney stiffness increased in patients with obesity, showing a statistically significant difference [15].

The correlations between fat mass and ultrasound elastography in each group are presented in Table 2 and in Figure 1, Figure 2 and Figure 3. Liver elastography correlated with all BC parameters but the least significantly with fat mass, which was the only parameter in BC that correlated significantly with kidney elastography. Liver elastography was the least dependent on fat mass, especially in healthy participants, where the correlation is negative; however, increasing stiffness in both kidneys was associated with increasing fat mass in patients with HTN and in the healthy individuals. 

A comparison among divided overweight/obese and normal-weight groups is presented in Table 3, and correlations between elastography and fat mass in each group in Figure 4, Figure 5 and Figure 6. As expected, groups differed according to anthropometric parameters due to the selection of the participants. In addition, systolic blood pressure was significantly elevated in participants with obesity. This was not true for diastolic blood pressure. In this part of the study, both liver and kidney elastography results were significantly higher in the overweight/obese group indicating an increase in stiffness with increasing body mass index. In addition, increasing kidney stiffness was associated with increasing fat mass also in healthy controls, in contrast with liver stiffness, which increased with fat mass only in overweight/obese participants.

Finally, a multiple regression analysis was performed among available variables—age, anthropometric measurements with BMI, systolic and diastolic pressure, BC parameters (PA, FM, FFM, BCM, TBW, and ECW)—presented in the Table 4. Models reached statistical significance in the F-test for regression equations, which, however, explained less than a third of the model. In addition, in liver elastography, only systolic pressure reached statistical significance as independent variable. In the left kidney elastography, height, hip circumference, and BCM affected the model; meanwhile, in the right kidney elastography, height and BCM were also significant, along with the PA and FFM. 

## 4. Discussion

Liver and kidney elastography are significantly affected by the presence of obesity [15], allowing further exploration of the association between elasticity parameters and BC. Results in the present study show an association between liver elastography and all BC parameters. Out of all, the correlation is the weakest between fat mass and liver elastography; however, on the contrary, fat mass is the only parameter associated with kidney elastography.

Despite age and height being consistent between groups, significant differences in BC between them are evident. The group with healthy subjects had no obese participants due to subject selection; however, both research groups had overweight/obese patients included. Even more, out of all hypertensive children, 76% were overweight/obese, which is not that surprising given the BMIs’ effect on blood pressure and increasing obesity-related HTN [18]. BC parameters were significantly elevated in both research groups, except for the amount of body fat, which was significantly higher only in the group with HTN, where obesity was more pronounced. It is important to point out that both research groups included patients with mild kidney disease and mild HTN, with the intent to study potential early changes in observed parameters.

Similar changes were seen after BMI division to the overweight/obese group and normal-weight group, where anthropometric differences became more prominent, as well as the differences between BC and UE measurements. The same was observed for systolic blood pressure, as expected according to previous research.

Increased liver stiffness parameters were measured in all patients compared to the healthy individuals and were even more pronounced after BMI division, suggesting that BMI significantly affects liver elasticity, which has already been shown in some studies [5,19]. In obesity, fat mass is responsible for excess weight; therefore, liver elastography parameters were expected to correlate well with fat mass, which was not entirely the case in the current study. Interestingly, liver UE correlated best with all other parameters but the least with fat mass, despite the fact that liver elastography parameters were elevated in all patients and were further aggravated by obesity. In vitro studies showed that liver stiffness increases depending on fat accumulation [20]; however, the extent of increasing liver stiffness with regard to the amount of adipose tissue has yet to be elucidated, both in in vivo conditions and separately in children, to discuss this phenomenon further. 

Although not reaching statistical significance, correlations within each group indicated an expected trend, which is even more pronounced after BMI division, where lean participants exhibited significant negative correlation. Interestingly, in a similar study, they found that the amount of adipose tissue measured by densitometry was also unrelated to the measurement of liver elasticity itself; however, it was related to other results of transient elastography [21]. Regardless of these inconclusive results that require further investigation, reduced liver elasticity showed an association with MAFLD in overweight/obese participants [5,19] with studies confirming the benefit of liver UE when following up patients with MAFLD among other indications [19,22]. 

Patients and healthy participants had similar kidney elastography results, in contrast to some other studies, where kidney UE differentiated in higher values of stiffness in scarred kidney areas [23] and detected an increase in some stiffness parameters in advanced CKD [24]. This might be due to the inclusion of patients with only mildly affected kidney function in the current study aiming to find an early kidney damage marker; however, the difference in kidney elastography became evident after BMI division, indicating an important effect of fat mass on kidney elasticity. Increased cortical kidney stiffness in overweight children has already been demonstrated [12]; however, it is clear from the parameters of BC that, unlike the elasticity of the liver, which is affected by all parameters of BC, the reduced elasticity of the kidneys is mainly influenced by the amount of adipose tissue. The positive association was demonstrated in almost all research groups, except for patients with CKD. This is of even greater importance knowing that excess weight is closely associated with renal damage [12], and, therefore, the early diagnosis of adverse renal effects of fat mass is essential for the prevention of progressive renal damage. 

Renal cortical stiffness was significantly higher in the overweight/obese group; however, in all groups in this study, fat mass correlated with left and right kidney elasticity (Figure 5 and Figure 6), demonstrating that, even in the normal-weight participants, the amount of fat mass is associated with kidney tissue elasticity. 

To the best of our knowledge and extensive research, there are limited studies researching the association between BC and ultrasound elastography in children. In adults, a similar study evaluated the association between liver and dual-energy X-ray absorptiometry. The mentioned study showed a similar positive correlation between BC and higher liver stiffness measured with transient elastography in the general population [21]. 

Otherwise, the data on fat mass’ effect on UE are lacking. In ultrasound imaging, fatty tissues produce acoustic attenuation and dispersion [25], adding an additional challenge in obtaining an appropriate image. Abdominal obesity was even associated with a decreased level of confidence in liver stiffness measurements [26]. However, a detailed procedure showed no correlations between composition parameters and elastography errors in adults [27], indicating a valid result when appropriate quality measures are achieved. In obesity, good imaging and quality control requires more time [28], which was also the case in the present study.

The strength of this study is a different angle in evaluating BIA and UE in children as potential tools for the early detection of cardiovascular- and/or obesity-associated damage beyond already known applications of both methods. It also indicates the importance of accurate fat mass measurement and its effect in monitoring the kidneys’ state. 

The main disadvantage of this study is the low number of subjects, which reduces the statistical power, especially when it is required to investigate the subjects according to the presence of obesity within each research group. The potential confounders influencing tissue elasticity are also numerous and could significantly affect our results. The group of children with CKD could also be improved, where the aetiology of the disease was very diverse. This helps us to evaluate the usefulness of UE regardless of the aetiology of the disease. With that said, the latter is not negligible, and more homogeneous groups would be required to define the influence of the aetiology of CKD on UE. Among hypertensive patients, not all were obese, and a group of normotensive children with obesity would be welcomed, which could provide additional insight in the dynamics between fat mass and UE. 

Several considerations need to take place when evaluating UE in the paediatric population. Growth and development are the cornerstone of childhood and are affecting many aspects of the physiologic state, influencing both UE and BC. Some studies, therefore, identified age to be an important influencing factor [29]; meanwhile, others found no differences between measurements taken at different ages and found no influence of the measurement depth [30,31] in contrast with adults [32]. In addition, gender did not significantly affect UE [5]. The limitations rather lie within the inadequate standardisation of UE [33]. Consequently, important factors, such as the type of ultrasound and probe, and operator- and patient-related factors are also potential causes of discrepancies and inconsistencies between studies [28]. 

Some of UE’s weaknesses can be surpassed using magnetic resonance elastography (MRE), which is another imaging technique used to evaluate tissue stiffness in clinical practice, most commonly to evaluate liver elasticity. MRE is not operator-dependent and not affected by obesity, which makes it more accurate in certain clinical settings. It can also measure the elasticity of the liver as a whole, rather than in a designated area [34]. MRE was found to be more accurate for staging liver fibrosis [34]. On the other hand, in patients with MAFLD, MRE and UE demonstrated a positive correlation [35], indicating either of them could be used for monitoring the disease progression. Interestingly, in the same study, neither MRE nor UE correlated with liver fat quantification [35], as was similarly demonstrated in the present study, where the correlations were present with borderline significance. The disadvantages of MRE include limited accessibility due to the appropriate hardware needed, which limits its use in clinical practice. The higher price of MRE hardware and software is another limiting factor.

BIA is known to be a practical, non-invasive method of measuring the percentage of fat mass [36], which is preferred in children. When used simultaneously with BMI’s percentiles, it showed improved sensitivity and specificity in identifying high-risk children who might benefit from appropriate management [37]. However, its validity and measurement error were not always satisfactory [36] with varying accuracy between BIA machines in obese or severely obese children [13]. Interestingly, the agreement between dual-energy X-ray absorptiometry and BIA was better for children with severe obesity than for children with mild to moderate obesity [38]. The most common reason for the discrepancies is the fact that all methods of BC measurement are “indirect” methods based on assumptions (using predictive equations) that might not be true in all situations or individuals, especially in children [39,40]. Balancing BIA’s advantages (accuracy, reproducibility, and ease of measure), it is still the preferred method for BC measurement [41], showing satisfying accuracy in some other studies [38,39,40,41,42], as well as performing well in measuring the change in BC in children [43]. Additionally, a systematic review showed BIA’s good reproducibility and correlation with the reference methods; however, fat mass was underestimated in both boys and girls [14], which is another concern when evaluating results.

Nevertheless, BIA was superior in the diagnosis of MAFLD among adults with a healthy BMI but elevated body fat [44], showing the potential importance of both UE as well as BIA in the investigated groups of children with mild pathology for the early detection of cardiovascular- and obesity-associated damage. Furthermore, the current study showed the effect of fat mass on kidney elasticity, emphasising the need for monitoring the kidney state in obese children with an accurate measurement of fat mass (cause) and its effect on kidney elastography (consequence). 

## 5. Conclusions

BC and fat mass are significantly associated with liver and kidney UE in the paediatric population. In this study, liver elastography significantly correlated with all BC parameters but the least with fat mass, which was the only BC parameter associated with kidney elastography. Our hypothesis was therefore confirmed—both liver and kidney elastography were associated with fat mass; however, this fact was especially pronounced in the kidneys and less in the liver, the opposite of what was expected. Further research is needed to clarify the effect of fat mass on tissue elasticity.

## Figures and Tables

**Figure 1 children-11-00018-f001:**
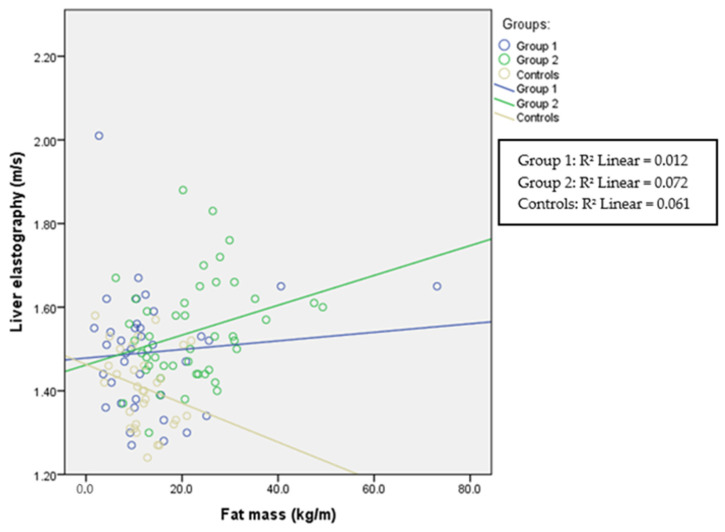
Liver elastography and fat mass in all three groups. Spearman’s correlation coefficient for Group 1 is r = 0.004 (*p* = 0.980), for Group 2 r = 0.273 (*p* = 0.066), and for the control group r = −0.250 (*p* = 0.176).

**Figure 2 children-11-00018-f002:**
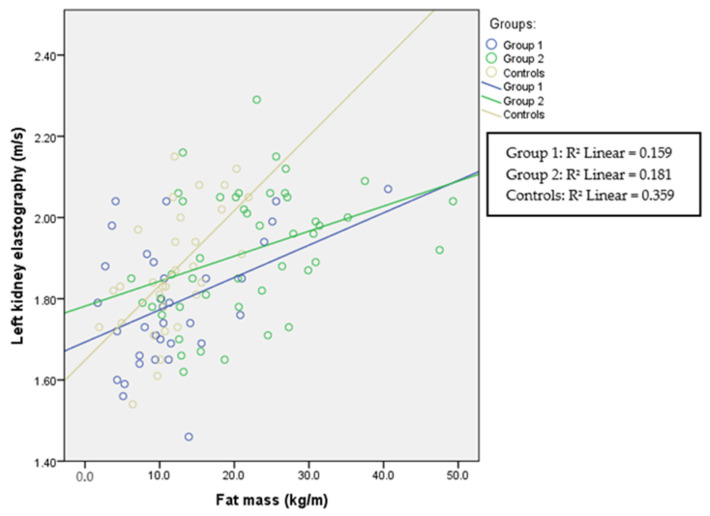
Left kidney elastography and fat mass. Spearman’s correlation coefficient for Group 1 is r = 0.266 (*p* = 0.129), for Group 2 r = 0.400 (*p* = 0.006), and for the control group r = 0.632 (*p* < 0.001).

**Figure 3 children-11-00018-f003:**
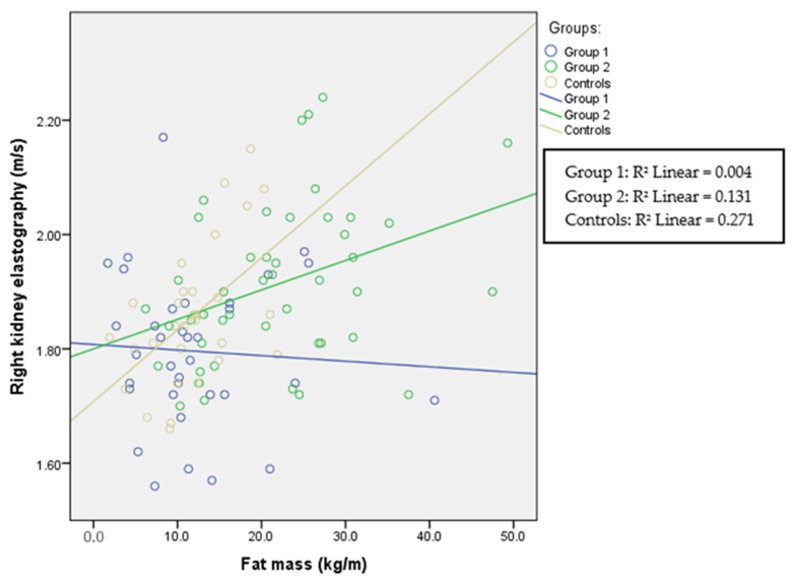
Right kidney elastography and fat mass. Spearman’s correlation coefficient for Group 1 is r = −0.112 (*p* = 0.522), for Group 2 r = 0.352 (*p* = 0.018), and for the control group r = 0.484 (*p* = 0.006).

**Figure 4 children-11-00018-f004:**
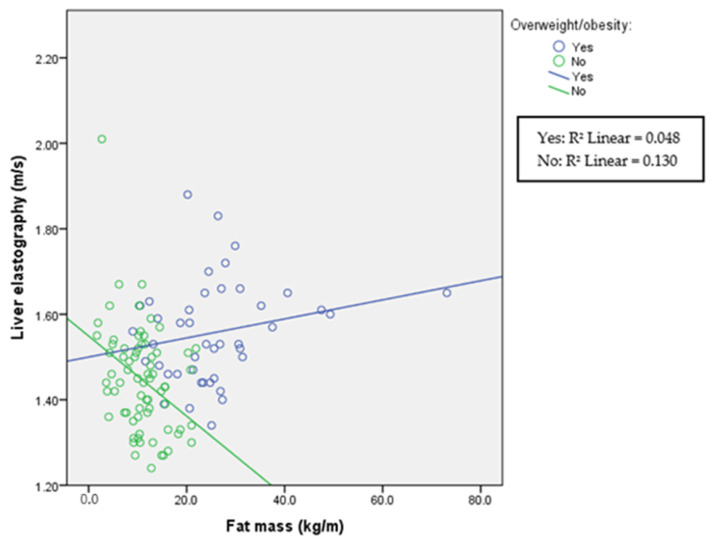
Liver elastography and fat mass according to overweight/obesity status. Spearman’s correlation coefficient for overweight participants is r = 0.234 (*p* = 0.130) and for normal-weight r = −0.306 (*p* = 0.009).

**Figure 5 children-11-00018-f005:**
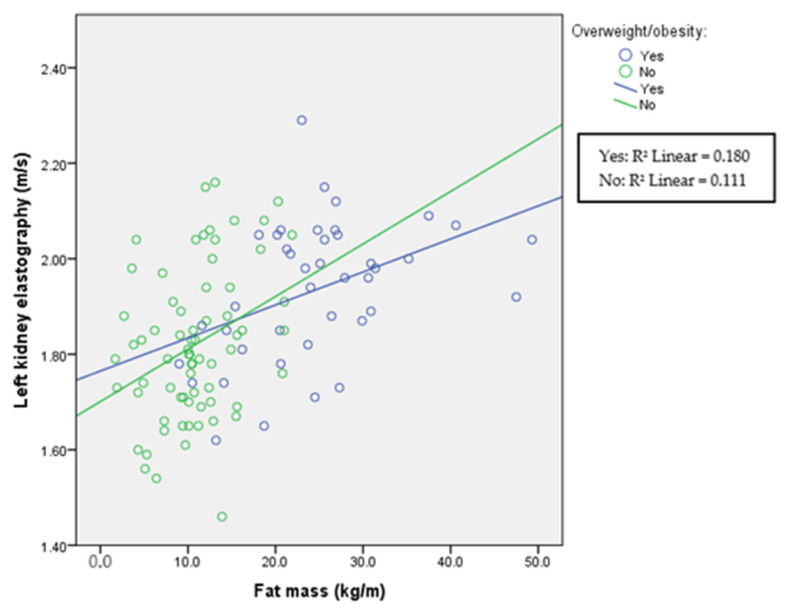
Left kidney elastography and fat mass according to overweight/obesity status. Spearman’s correlation coefficient for overweight participants is r = 0.433 (*p* = 0.005) and for normal-weight r = 0.331 (*p* = 0.005).

**Figure 6 children-11-00018-f006:**
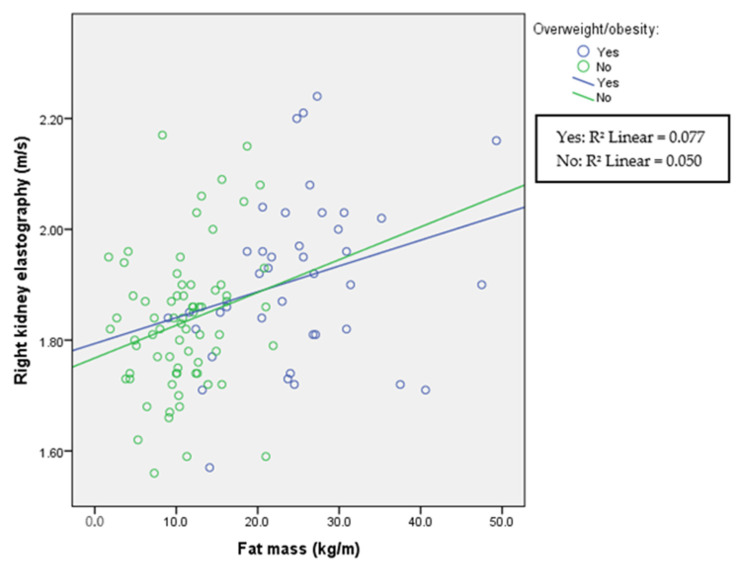
Right kidney elastography and fat mass according to overweight/obesity status. Spearman’s correlation coefficient for overweight participants is r = 0.277 (*p* = 0.083) and for normal-weight r = 0.234 (*p* = 0.050).

**Table 1 children-11-00018-t001:** Groups’ variables, presented as median (interquartile range) or numbers (relative percentages) with comparison to control group using Mann–Whitney test. Additionally, both research groups were compared (fifth column) and all three groups were evaluated using Kruskal–Wallis test.

Variable	Group 1 (N = 37) “*CKD Group*”	Group 2 (N = 46) “*HTN Group*”	Healthy Subjects (N = 31)	Group 1 vs. Group 2	Kruskal–Wallis Test
Age (years)	16 (6) *p* = 0.250	15 (6) *p* = 0.806	14 (4)	*p* = 0.364	*p* = 0.488
M/F	23/14	37/9	13/18		
Height (cm)	170 (16.3) *p* = 0.108	169 (22.5) *p* = 0.085	163 (16)	*p* = 0.780	*p* = 0.165
Weight (kg)	60 (21.90.146) *p* = 0.146	76.8 (27.8) *p* < 0.001	53 (19.5)	*p* = 0.001	*p* < 0.001
BMI (kg/m^2^)	20.8 (4.7) *p* = 0.252	26.5 (6) *p* < 0.001	19.9 (3)	*p* < 0.001	*p* < 0.001
Overweight/obese	8 (21.6%)	35 (76%)	0		
Waist circumference (cm)	73 (10) *p* = 0.250	89 (15) *p* < 0.001	70 (9)	*p* < 0.001	*p* < 0.001
Hip circumference (cm)	83.5 (17) *p* = 0.792	96 (17) *p* < 0.001	85 (10)	*p* = 0.001	*p* < 0.001
Systolic pressure (mmHg)	119 (11) *p* = 0.032	130 (11) *p* < 0.001	116 (15)	*p* < 0.001	*p* < 0.001
Diastolic pressure (mmHg)	70 (13) *p* = 0.878	71 (10) *p* = 0.735	69 (10)	*p* = 0.660	*p* = 0.891
LE (m/s)	1.51 (0.18) *p* = 0.008	1.52 (0.16) *p* < 0.001	1.41 (0.18)	*p* = 0.125	*p* < 0.001
LE (kPa)	6.7 (1.7) *p* = 0.007	6.7 (1.6) *p* < 0.001	5.9 (1.5)	*p* = 0.176	*p* < 0.001
Left KE (m/s) *	1.77 (0.21) *p* = 0.041	1.9 (0.26) *p* = 0.224	1.84 (0.26)	*p* = 0.001	*p* = 0.003
Left KE (kPa) *	9.35 (2.23) *p* = 0.058	11.1 (3) *p* = 0.156	10.1 (2.9)	*p* = 0.001	*p* = 0.002
Right KE (m/s)	1.79 (0.16) *p* = 0.074	1.9 (0.21) *p* = 0.079	1.85 (0.11)	*p* = 0.001	*p* = 0.002
Right KE (kPa)	9.5 (1.7) *p* = 0.084	10.9 (2.5) *p* = 0.073	10.3 (1.1)	*p* = 0.001	*p* = 0.002
PA (°)	6.7 (1.3)0.003 *p* = 0.003	7.0 (1.4) *p* < 0.001	6 (1)	*p* = 0.190	*p* < 0.001
FM (kg/m)	10.5 (8.6) *p* = 0.631	20.9 (14.1) *p* < 0.001	11.8 (5.7)	*p* < 0.001	*p* < 0.001
FFM (kg/m)	48.2 (19.1) *p* = 0.016	55.6 (18.8) *p* < 0.001	40.9 (10.7)	*p* = 0.086	*p* < 0.001
BCM (kg/m)	27.6 (15.5) *p* = 0.047	32.8 (12.1) *p* < 0.001	22.8 (6.8)	*p* = 0.080	*p* < 0.001
TBW (L/m)	37.4 (14.5) *p* = 0.027	41.9 (13.1) *p* < 0.001	32.6 (7.2)	*p* = 0.082	*p* < 0.001
ECW (L/m)	16.3 (4.9) *p* = 0.021	17.2 (6.2) *p* = 0.001	13.8 (4)	*p* = 0.122	*p* = 0.002

CKD—chronic kidney disease, HTN—hypertension, M/F—number of male and female participants, BMI—body mass index, LE—liver elastography, KE—kidney elastography, PA—phase angle, FM—fat mass, FFM—fat-free mass, BCM—body cell mass, TBW—total body water, ECW—extracellular water. * Elastography measures the movement of tissue, expressed as m/s that is further computed to elasticity model, expressed as kPa. Values are, therefore, collinear.

**Table 2 children-11-00018-t002:** Correlations between liver and kidney elastography and body composition in all participants using Spearman’s correlation coefficient.

	LE (m/s)	LE (kPa)	Left KE (m/s)	Left KE (kPa)	Right KE (m/s)	Right KE (kPa)
PA (°)	r = 0.284 *p* = 0.002	r = 0.277 *p* = 0.003	r = 0.019 *p* = 0.845	r = 0.045 *p* = 0.636	r = −0.024 *p* = 0.801	r = −0.001 *p* = 0.995
FM (kg/m)	r = 0.191 *p* = 0.041	r = 0.178 *p* = 0.058	r = 0.514 *p* < 0.001	r = 0.541 *p* < 0.001	r = 0.367 *p* < 0.001	r = 0.392 *p* < 0.001
FFM (kg/m)	r = 0.294 *p* = 0.001	r = 0.288 *p* = 0.002	r = −0.032 *p* = 0.739	r = 0.005 *p* = 0.961	r < 0.001 *p* = 0.998	r = 0.027 *p* = 0.781
BCM (kg/m)	r = 0.303 *p* = 0.001	r = 0.296 *p* = 0.001	r = −0.023 *p* = 0.814	r = 0.013 *p* = 0.892	r = −0.006 *p* = 0.947	r = 0.021 *p* = 0.829
TBW (L/m)	r = 0.307 *p* = 0.001	r = 0.299 *p* = 0.001	r = −0.060 *p* = 0.535	r = −0.022 *p* = 0.819	r = −0.017 *p* = 0.860	r = 0.008 *p* = 0.932
ECW (L/m)	r = 0.254 *p* = 0.006	r = 0.246 *p* = 0.008	r = −0.103 *p* = 0.282	r = −0.069 *p* = 0.470	r = −0.066 *p* = 0.491	r = −0.043 *p* = 0.654

LE—liver elastography, KE—kidney elastography, PA—phase angle, FM—fat mass, FFM—fat-free mass, BCM—body cell mass, TBW—total body water, ECW—extracellular water.

**Table 3 children-11-00018-t003:** Groups’ variables, presented as median (interquartile range) or numbers (relative percentages) with comparison between overweight and normal-weight group using Mann–Whitney test.

Variable	Group with Obesity (N = 43)	Group with Normal Weight (N = 71)	Comparison
Age (years)	15 (7)	15 (5)	*p* = 0.649
M/F	35/8	38/33	
Height (cm)	169 (17)	168 (17)	*p* = 0.684
Weight (kg)	80 (27)	59 (18.5)	*p* < 0.001
BMI (kg/m^2^)	27.9 (5.5)	20 (3.2)	*p* < 0.001
Waist circumference (cm)	91.5 (14)	72 (11)	*p* < 0.001
Hip circumference (cm)	99 (14)	84 (11)	*p* < 0.001
Systolic pressure (mmHg)	129 (14)	118 (17)	*p* < 0.001
Diastolic pressure (mmHg)	70 (10)	70 (11)	*p* = 0.619
LE (m/s)	1.53 (0.17)	1.44 (0.16)	*p* < 0.001
LE (kPa)	6.9 (1.6)	6.1 (1.4)	*p* < 0.001
Left KE (m/s)	1.96 (0.21)	1.81 (0.21)	*p* < 0.001
Left KE (kPa)	11.8 (2.5)	9.8 (2.4)	*p* < 0.001
Right KE (m/s)	1.91 (0.2)	1.83 (0.15)	*p* = 0.007
Right KE (kPa)	11.1 (2.3)	10 (1.6)	*p* = 0.003
PA (°)	7 (1.4)	6.3 (1.2)	*p* < 0.001
FM (kg/m)	24.5 (11.2)	10.6 (5.4)	*p* < 0.001
FFM (kg/m)	58.6 (21.9)	45 (15.5)	*p* < 0.001
BCM (kg/m)	33.4 (15.1)	10.6 (5.4)	*p* < 0.001
TBW (L/m)	42.5 (14.4)	33.7 (11.2)	*p* < 0.001
ECW (L/m)	17.2 (6.3)	15.1 (5)	*p* = 0.006

M/F—number of male and female participants, BMI—body mass index, LE—liver elastography, KE—kidney elastography, PA—phase angle, FM—fat mass, FFM—fat-free mass, BCM—body cell mass, TBW—total body water, ECW—extracellular water.

**Table 4 children-11-00018-t004:** Multiple regression analysis results for liver and kidney elastography as dependent variables.

	Liver Elastography	Left Kidney Elastography	Right Kidney Elastography
Age (years)	t = −0.520, *p* = 0.604	t = 1.770, *p* = 0.080	t = 0.248, *p* = 0.805
Height (cm)	t = −1.807, *p* = 0.074	t = −2.069, *p* = 0.041	t = −2.526, *p* = 0.013
Weight (kg)	t = −0.519, *p* = 0.605	t = 1.497, *p* = 0.138	t = 1.504, *p* = 0.136
BMI (kg/m^2^)	t = −1.374, *p* = 0.173	t = −1.453, *p* = 0.150	t = −1.060, *p* = 0.292
Waist circumference (cm)	t = 1.915, *p* = 0.059	t = 1.173, *p* = 0.244	t = −0.236, *p* = 0.814
Hip circumference (cm)	t = −1.675, *p* = 0.097	t = 2.106, *p* = 0.038	t = 0.913, *p* = 0.364
Systolic pressure (mmHg)	t = 2.529, *p* = 0.013	t = 0.999, *p* = 0.321	t = 1.376, *p* = 0.172
Diastolic pressure (mmHg)	t = −1.208, *p* = 0.230	t = −0.220, *p* = 0.826	t = −1.539, *p* = 0.127
PA (°)	t = 1.078, *p* = 0.284	t = 1.949, *p* = 0.054	t = 2.535, *p* = 0.013
FM (kg/m)	t = 1.047, *p* = 0.298	t = −0.842, *p* = 0.402	t = −0.893, *p* = 0.374
FFM (kg/m)	t = 1.130, *p* = 0.261	t = 1.147, *p* = 0.254	t = 2.387, *p* = 0.019
BCM (kg/m)	t = −1.082, *p* = 0.282	t = −2.272, *p* = 0.025	t = −3.030, *p* = 0.003
TBW (L/m)	t = 2.005, *p* = 0.048	t = 0.308, *p* = 0.759	t = −0.807, *p* = 0.422
ECW (L/m)	t = −1.066, *p* = 0.289	t = −1.225, *p* = 0.224	t = −1.137, *p* = 0.259
F-test	3.083, *p* = 0.001	4.652, *p* < 0.001	2.983, *p* = 0.001
R^2^	0.315	0.414	0.312
Adjusted R^2^	0.213	0.325	0.208

BMI—body mass index, PA—phase angle, FM—fat mass, FFM—fat-free mass, BCM—body cell mass, TBW—total body water, ECW—extracellular water.

## Data Availability

The data presented in this study are available upon request from the corresponding author. The data are not publicly available due to privacy and ethical restrictions.

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
