# Peer review of "Obesity and Body Composition in Relation to Liver and Kidney Ultrasound Elastography in Paediatric Patients with Either Hypertension or Chronic Kidney Disease"

_children, 2023, doi:10.3390/children11010018_

Round 1
Reviewer 1 Report
Comments and Suggestions for Authors
The purpose of this manuscript is to assess the relationship between obesity, body composition parameters, and liver and kidney ultrasound elastography in paediatric patients. I think it is interesting and useful to explore the relationship between obesity and ultrasound elastography.
In my opinion, the current version of the manuscript is not suitable for publication unless a more thorough analysis or additional works are provided. The main reasons for this recommendation are as follows.
1. The authors' definition of control group is inherently wrong. The authors' groupings are in fact subgroups or stratified analyses for the dependent variable of the study.
2. All of the authors' analyses are based on simple correlation analysis. I believe this is completely inadequate and, due to the lack of multifactorial regression analysis, potential confounders may have affected the results of the study. For example, the distribution of potential confounders between obese and normal groups may be uneven, especially in small sample size studies.
3. Therefore, I suggest that the authors redefine the so-called control group, incorporate the necessary covariates, and supplement the multifactorial analysis. On this basis, the independent association between obesity and UE should be explored in more depth.
Other minor issues:
1. The testing procedures for all variables should be more detailed rather than referring to published articles. Obesity cut points should be noted. All data collected can be duplicated by others.
2. Abbreviations should be labeled below the table, and also, the statistical method for p-values should be stated.
3. the R and R2 value need to be checked.
Author Response
“The purpose of this manuscript is to assess the relationship between obesity, body composition parameters, and liver and kidney ultrasound elastography in paediatric patients. I think it is interesting and useful to explore the relationship between obesity and ultrasound elastography.
In my opinion, the current version of the manuscript is not suitable for publication unless a more thorough analysis or additional works are provided. The main reasons for this recommendation are as follows.”
“1. The authors' definition of control group is inherently wrong. The authors' groupings are in fact subgroups or stratified analyses for the dependent variable of the study.” – This is true for our grouping according to obesity status; however, the basic control group consisted of only healthy individuals invited separately. This means that in the normal-weight group there were indeed some patients with either chronic kidney disease or hypertension and all of healthy participants (the control group). Due to your well-put comment, additional clarification of grouping has been added in several parts of the manuscript.
“2. All of the authors' analyses are based on simple correlation analysis. I believe this is completely inadequate and, due to the lack of multifactorial regression analysis, potential confounders may have affected the results of the study. For example, the distribution of potential confounders between obese and normal groups may be uneven, especially in small sample size studies.” – Agreed. On the other hand, this is also the reason this study is interesting, because the differences in elasticity according to fat mass, were significantly different in already small group, where the effect of other confounders would be greater. In our study we intentionally focused only on body composition (with emphasis on fat mass) to evaluate its effect only above standard anthropometric measurements. Due to your well-put comment, additional multiple regression analysis was performed using available variables and the study weakness has been additionally clarified.
“3. Therefore, I suggest that the authors redefine the so-called control group, incorporate the necessary covariates, and supplement the multifactorial analysis. On this basis, the independent association between obesity and UE should be explored in more depth.” – Answered above.
“Other minor issues:
- The testing procedures for all variables should be more detailed rather than referring to published articles. Obesity cut points should be noted. All data collected can be duplicated by others.” – Further details to the testing procedures have been added. Obesity cut points were derived from another source and were not noted to avoid the repetition. The reference is stated.
“2. Abbreviations should be labeled below the table, and also, the statistical method for p-values should be stated.” – Thank you. Abbreviations were moved and statistical method added.
“3. the R and R2 value need to be checked.” – Checked additionally, it is what SPSS computes. The graphs are copied from the computer software.
– Thank you for your comments and thank you very much for taking your time in improving our manuscript.
Reviewer 2 Report
Comments and Suggestions for Authors
-
The study sought to assess body composition parameters with kidney and liver ultrasound elastography in pediatric patients. The manuscript was well-organized, but not well-written; however, several modifications are required.
· -The discussion part did have good quality and there was a major weakness in that part. The discussion should have more references to compare the outcomes with other previous studies.
· Please add a statement regarding the strengths of the study before limitation and add more limitations.
· The rationale of the study should be explained at the end of the introduction.
· The readability of the results section should be improved.
· The words "our" and "we" should be removed from the manuscript and replaced with other words such as "current study", "this study," and "present study."
· Extensive paraphrasing and proofreading are required.
- Extensive paraphrasing and proofreading are required.
Author Response
To Reviewer 2:
“The study sought to assess body composition parameters with kidney and liver ultrasound
elastography in pediatric patients. The manuscript was well-organized, but not well-written;
however, several modifications are required.
- -The discussion part did have good quality and there was a major weakness in
that part. The discussion should have more references to compare the outcomes with
other previous studies.” – The discussion part has been rewritten in some parts and some references were added; however, to our knowledge and extensive research, there is scarce similar studies for comparison as already stated in the manuscript.
- “Please add a statement regarding the strengths of the study before limitation
and add more limitations.” – Added.
- “The rationale of the study should be explained at the end of the introduction.” – Added.
- “The readability of the results section should be improved.” – Rewritten and rephrased in some parts.
- “The words “our” and “we” should be removed from the manuscript and replaced with other words such as “current“, “this“ and “present study” – Corrected as suggested.
- “Extensive paraphrasing and proofreading are required.” – Addition rephrasing and proofreading were made.
– Thank you for your comments and thank you very much for taking your time in improving our manuscript.
Reviewer 3 Report
Comments and Suggestions for Authors
It is an interesting work that contributes to knowledge about the usefulness of ultrasound elastography in mild conditions and in subjects with obesity. Some observations on the manuscript.
Abstract.
The aim in the Abstract (evaluate body composition parameters with liver and kidney ultrasound elastography in pediatric patients) is not the same as that of the text (evaluate the effect of body composition and fat mass on liver and kidney UE in pediatric patients with either hypertension or chronic kidney disease).
Methods.
Although authors refer to a previous published article, please briefly describe the origin of the included patients and the characteristics of the equipment with which the ultrasound elastography was measured.
Results.
I suggest adding the results for liver and kidney stiffness in Table 1; the same ones that are written in lines 115-118.
Table 2. Add as a footnote to the table the meaning of m/s and kPa
Discussion.
I suggest adding to the discussion a justification of why increased liver stiffness shows a barely significant association with fat mass; but it was significantly higher in subjects with obesity. ¿Are there other mechanisms involved in the development of greater liver stiffness in overweight subjects in addition to excess body fat?
Add the strengths or weaknesses of Ultrasound Elastography with respect to other techniques for evaluating liver stiffness and/or fatty liver disease. Price, equipment availability, accessibility.
Author Response
To Reviewer 3:
“It is an interesting work that contributes to knowledge about the usefulness of ultrasound elastography in mild conditions and in subjects with obesity. Some observations on the manuscript.
Abstract.
The aim in the Abstract (evaluate body composition parameters with liver and kidney ultrasound elastography in pediatric patients) is not the same as that of the text (evaluate the effect of body composition and fat mass on liver and kidney UE in pediatric patients with either hypertension or chronic kidney disease).” – Rephrased.
“Methods.
Although authors refer to a previous published article, please briefly describe the origin of the included patients and the characteristics of the equipment with which the ultrasound elastography was measured.” – Added.
“Results.
I suggest adding the results for liver and kidney stiffness in Table 1; the same ones that are written in lines 115-118.” – Added.
“Table 2. Add as a footnote to the table the meaning of m/s and kPa” – Added.
“Discussion.
I suggest adding to the discussion a justification of why increased liver stiffness shows a barely significant association with fat mass; but it was significantly higher in subjects with obesity. Are there other mechanisms involved in the development of greater liver stiffness in overweight subjects in addition to excess body fat? “ – This is an excellent notion and an additional comment has been added in the manuscript. A reliable answer supported by references cannot be provided at the moment, but the phenomenon will be followed further. It might be due to comparison of an absolute value to relative, which correlations are – the increasing liver stiffness after a certain point might not be linear. In a study, performed in vivo in adults and also in vitro, there is clear notion, that stiffness increases as a function of extent of fat accumulation (https://www.ncbi.nlm.nih.gov/pmc/articles/PMC9139073/), however in graphs (Figure 4A) it is seen that relative elasticity does not function linear between controls, moderate and severe steatosis. Interestingly, in other studies using also MRE along with UE, similarly correlations with fat mass were limited (https://journals.lww.com/ultrasound-quarterly/abstract/2023/06000/comparison_of_magnetic_resonance_based.6.aspx).
“Add the strengths or weaknesses of Ultrasound Elastography with respect to other techniques for evaluating liver stiffness and/or fatty liver disease. Price, equipment availability, accessibility.” – Added.
– Thank you for your comments and thank you very much for taking your time in improving our manuscript.
Reviewer 4 Report
Comments and Suggestions for Authors
Though this study overall is rather of a confirmatory nature, it is interesting that particualrly kidney elastography correlated significantly with fat mass, while, as expected, liver UE correlated with all body composition parameters but to a lesser extent with fat mass.
The study is well conducted and presented. However, I´d wish the authors could better explain the aims of the study and the possible impact of their results.
Comments on the Quality of English LanguageI am not qualified to correct the language of the paper but at some points, particularly in the discussion, it was difficult to understand what the authors exactly wanted to express. I would suggest to have the text reviewed by an expert in scientific English.
Author Response
To Reviewer 4:
“Though this study overall is rather of a confirmatory nature, it is interesting that particularly kidney elastography correlated significantly with fat mass, while, as expected, liver UE correlated with all body composition parameters but to a lesser extent with fat mass.
The study is well conducted and presented. However, I´d wish the authors could better explain the aims of the study and the possible impact of their results.” – Additionally rewritten and explained in several parts of the manuscript.
“Comments on the Quality of English Language
I am not qualified to correct the language of the paper but at some points, particularly in the discussion, it was difficult to understand what the authors exactly wanted to express. I would suggest to have the text reviewed by an expert in scientific English.” – An expert in medical English has revised the manuscript. Also, the manuscript has been rewritten in some parts.
– Thank you for your comments and thank you very much for taking your time in improving our manuscript.
Round 2
Reviewer 1 Report
Comments and Suggestions for Authors
1. the authors may not properly understand the meaning of control group. Your study design is not an experimental study. All groups in Tables 1 and 3 should have been used with their original names, especially the so-called control group, which should have been deleted and replaced with a healthy group.
2. Multiple regression should include all independent variables, presented in Table 4, and be supplemented with the following tests: t-tests for independent variables, F-tests for regression equations, and goodness-of-fit tests (R2). Waist and hip circumference, which seem to have multicollinearity, should also be tested.
Author Response
»1. the authors may not properly understand the meaning of control group. Your study design is not an experimental study. All groups in Tables 1 and 3 should have been used with their original names, especially the so-called control group, which should have been deleted and replaced with a healthy group.« – Thank you for the clarification. Noted and corrected as suggested.
»2. Multiple regression should include all independent variables, presented in Table 4, and be supplemented with the following tests: t-tests for independent variables, F-tests for regression equations, and goodness-of-fit tests (R2). Waist and hip circumference, which seem to have multicollinearity, should also be tested.« – Noted and corrected as suggested. This, therefore, is not a stepwise approach. Table 4 has been changed accordingly.
– Thank you for your comments and thank you very much for taking your time in improving our manuscript.
Reviewer 2 Report
Comments and Suggestions for Authors
The authors tried to modify the manuscript based on suggestion.
The word “our” should be removed from the manuscript; lines 200,262,284, 66,67, 10.
Comments on the Quality of English Language
Minor proofreading is required.
Author Response
To Reviewer 2:
»The authors tried to modify the manuscript based on suggestion.
The word “our” should be removed from the manuscript; lines 200,262,284, 66,67, 10.« – Removed.
– Thank you for your comments and thank you very much for taking your time in improving our manuscript.
Reviewer 4 Report
Comments and Suggestions for Authors
This study shows that liver and kidney ultrasound elastorgraphy (UE) results depend on body composition and fat mass in a pediatric population. In adults, similar relations between body composition and higher liver stiffness are described, while no or limited studies in children are published. IN this study on pediatric patients liver elastography significantly correlated with all body composition parameters but remarkably weak with fat mass, which on the other hand was the only body composition parameter, associated with kidney elastography. Elucidating the effect of fat mass on tissue elasticity clearly demands further research, the data of this manuscript may lay some basis for that.
UE may have some advantages regarding accessibility and costs compared with MRE for monitoring tissue stiffness and thus risk factors in clinical practice and the results shown his study emphasizes the importance of monitoring particularly the kidney in pediatric patients. However, as clearly expressed by the authors in their discussion, the method is far from being standardized. It may be also a contribution of this study to promote such standardization.
Thus, despite the weaknesses that are openly discussed by the authors, the data of this study may be an important contribution to the field.
The issues addressed in the first review were adequately addressed. I´d have wished to find a version with the changes marked, but obviously, this is not required by the journal´s submission procedure.
Minor comment:
Please include a better legend to Table 1 so that it can be read independently of the text (explain G1 and G2, or better: give self-explanatory names ti this groups(applies also to GO1, GO2)).
Author Response
To Reviewer 4:
“This study shows that liver and kidney ultrasound elastorgraphy (UE) results depend on body composition and fat mass in a pediatric population. In adults, similar relations between body composition and higher liver stiffness are described, while no or limited studies in children are published. IN this study on pediatric patients liver elastography significantly correlated with all body composition parameters but remarkably weak with fat mass, which on the other hand was the only body composition parameter, associated with kidney elastography. Elucidating the effect of fat mass on tissue elasticity clearly demands further research, the data of this manuscript may lay some basis for that.
UE may have some advantages regarding accessibility and costs compared with MRE for monitoring tissue stiffness and thus risk factors in clinical practice and the results shown his study emphasizes the importance of monitoring particularly the kidney in pediatric patients. However, as clearly expressed by the authors in their discussion, the method is far from being standardized. It may be also a contribution of this study to promote such standardization.
Thus, despite the weaknesses that are openly discussed by the authors, the data of this study may be an important contribution to the field.
The issues addressed in the first review were adequately addressed. I´d have wished to find a version with the changes marked, but obviously, this is not required by the journal´s submission procedure.
Minor comment:
Please include a better legend to Table 1 so that it can be read independently of the text (explain G1 and G2, or better: give self-explanatory names ti this groups(applies also to GO1, GO2)).” – Corrected.
– Thank you for your comments and thank you very much for taking your time in improving our manuscript.